# Lucy: Think and Reason to Solve Text-to-SQL

**Submission #14824**

## Abstract

Large Language Models (LLMs) have made significant progress in assisting users to query databases in natural language. While LLM-based techniques provide state-of-the-art results on many standard benchmarks, their performance significantly drops when applied to large enterprise databases. The reason is that these databases have a large number of tables with complex relationships that are challenging for LLMs to reason about. We analyze challenges that LLMs face in these settings and propose a new solution that combines the power of LLMs in understanding questions with automated reasoning techniques to handle complex database constraints. Based on these ideas, we have developed a new framework that outperforms state-of-the-art techniques in zero-shot text-to-SQL on complex benchmarks.

## 1 Introduction

Large Language Models (LLMs) have significantly enhanced AI agents' capacity to assist humans in a variety of important tasks, including co-pilot programming [Chen et al., 2021, GitHub, Inc., 2021], program verification [Wu et al., 2024, Chakraborty et al., 2023], and math problem solving [Zhou et al., 2024]. One of the fastest-growing areas in this space is the development of LLM-based assistants for querying SQL databases. In this task, a user poses a question to a database in natural language. The agent's goal is to generate an SQL query that, when executed against the database, answers the user's question. Such assistance enables users with different levels of expertise to effectively analyze their data.

Recently, LLM-based solutions have made significant progress in addressing the text-to-SQL problem [Gao et al., 2024, Li et al., 2024a]. While GPT-based methods have quickly reached near-human performance on academic benchmarks, like Spider [Yu et al., 2018], they struggle to provide high-quality user assistance on large industrial databases [Sequeda et al., 2023, Li et al., 2023]. One of the core challenges is that industrial databases model many objects with complex relationships between them. To transform a natural language question into an SQL query, the LLM must effectively reason about these intricate relationships, which is highly non-trivial for LLM models. Interestingly, we found that GPT4 can even indicate in some cases that it needs help with logical reasoning on complex databases. Here is a common GPT4 output message on a question that requires multiple joins from ACME insurance database [Sequeda et al., 2023]: *'This join may need adjustment based on the actual logic of relating claims to policy coverage details.'*. While we do provide the database schema as part of the input, it is still challenging for LLMs to formally reason about database logic.

In this work, we propose a new text-to-SQL framework, Lucy, designed for large databases with complex relationships between objects. Our main underlying idea is to combine the ability of LLM models to effectively relate user questions to database objects with the power of automated reasoning to analyze relationships between these objects. The Lucy workflow consists of three high-level steps. First, upon receiving a user's question, we identify the relevant objects and their attributes in the target database. In the second step, we employ an automated reasoner to build a view that joins the relevant tables based on relational constraints defined by the database schema. This view contains all the necessary information

Submitted to 38th Conference on Neural Information Processing Systems (NeurIPS 2024). Do not distribute.

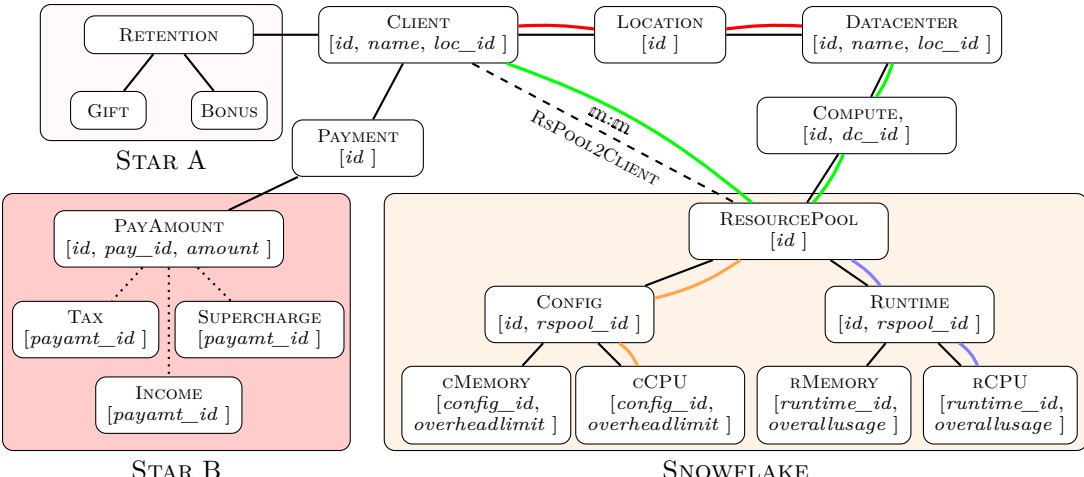

Figure 1: Objects and their relations in the database DDO.

to answer the user's questions. In the third step, we construct a query targeting this view to produce an answer for the user. Our contributions are summarized as follows:

- We propose a text-to-SQL framework LUCY capable of querying large industrial databases. To the best of our knowledge, LUCY is the first framework designed to support logical reasoning in the context of the text-to-SQL problem.
- LUCY offers several advantages:
  - alleviates the need for complex reasoning from a LLM, allowing it to focus on tasks where it currently excels,
  - supports modeling and reasoning about complex, commonly used design patterns to model relationships, like many-to-many, STAR, and SNOWFLAKE,
  - its modular workflow allows for effective debugging of failures,
  - performs zero-shot generation and does not require fine-tuning of LLMs.
- Our experimental results demonstrate significant performance improvements on several standard benchmarks as well as introduced large benchmarks. We also demonstrate the debugging capabilities of LUCY.

## 2  Motivation

To provide high-quality user assistance in text-to-SQL tasks, we face two types of challenges. The first type of challenge comes from the formulation of the user's question. A question can be poorly specified, ambiguous, or require additional knowledge that is not present in the question. For example, the user might ask to list clients eligible for a loan; however, the eligibility criteria are not present in the question [Li et al., 2023, 2024b]. The second class is related to the complexity of the queried database that can have a large number of tables with complex relations between them [Sequeda et al., 2023, Li et al., 2023]. In this work, we focus on the second class. One approach to deal with complex relationships is to introduce an intermediate layer, like a knowledge graph or ontology structure, that contains rich information about the underlying database. Then, LLMs generate queries to this knowledge graph using specialized languages, e.g., SPARQL, [Sequeda et al., 2023]. In turn, these queries can be automatically translated to SQL. While this approach does show promise, it does not alleviate the core issue: an LLM is still expected to reason about complex relations between objects in this intermediate representation. Moreover, such a rich intermediate layer, like an ontology, might not be easy to obtain for a database. Other standard techniques, like additional training, multi-shot or fine-tuning, also rely on LLMs to perform constrained reasoning steps [Gao et al., 2023, Pourreza and Rafiei, 2024, Gao et al., 2024]. To the best of our knowledge, dealing with complex relationships in text-to-SQL remains an open problem. In order to isolate the underlying challenges in this problem, we created an example database

**Table 1 (Q1):**

*Q1: List customers who use datacenters with names starting with 'dev'. Output clients and datacenters names.*

```
/*GPT4 generated SQL*/:
select Client.name, Datacenter.name
from Client
join Location on Location.id = Client.loc_id
join Datacenter on Location.id = Datacenter.loc_id
where Datacenter.name like 'dev%'
```

```
/*Correct SQL*/
select Client.name, Datacenter.name
from Datacenter
join Compute on Datacenter.id = Compute.dc_id
join ResourcePool on
    Compute.id = ResourcePool.compute_id
join RsPool2Client on
    ResourcePool.id = RsPool2Client.rspool_id
join Client on Client.id = RsPool2Client.client_id
where Datacenter.name like 'dev%'
```

**Table 1 (Q2):**

*Q2: List resource pools names with CPU overhead limit greater than runtime overall usage by 100.*

```
/*GPT4 generated SQL*/:
select ResourcePool.name
from ResourcePool
join rCPU on
    ResourcePool.runtime_id = rCPU.runtime_id
join cCPU on
    ResourcePool.config_id = cCPU.config_id
where cCPU.overheadlimit > rCPU.overallusage + 100
```

```
/*Correct SQL*/:
select distinct ResourcePool.name
from ResourcePool
left join Config on
    ResourcePool.id = Config.rspool_id
left join cCPU on Config.id = cCPU.config_id
left join Runtime on
    ResourcePool.id = Runtime.rspool_id
left join rCPU on Runtime.id = rCPU.runtime_id
where cCPU.overheadlimit > rCPU.overallusage + 100
```

Table 1: User's questions Q1 and Q2. Incorrect parts of the GPT answer are shown in red.

that covers standard relationship patterns adopted in industry and academia. We identified a set of simple and clearly formulated questions and demonstrated that even on this simplified schema and clear questions, state-of-the-art LLMs struggle to assist the user.

## 2.1 Database description

We describe a minimal example database schema that contains basic relations, like 1:1 and 1:m, and more advanced relationship patterns, like m:m and STAR, and analyze the performance of LLMs on this schema (See Appendix A for relational database definitions). Suppose a business sells cloud compute resources to customers and uses a database, DDO, to manage its Day-to-Day Operations. Figure 1 shows objects' corresponding tables, their relationships, and a subset of attributes. In particular, each table has a primary key, e.g., LOCATION.*id*, and might have foreign keys to refer to another table, e.g., CLIENT refers to LOCATION using CLIENT.*loc_id*. All attributes relevant to our examples are shown in Figure 1 with self-explanatory names. DDO manages payments (PAYMENT) and marketing retention strategies (RETENTION) for clients (CLIENT) and resources (RESOURCEPOOL) in datacenters (DATACENTER). This example is in part inspired by the VMware vSphere data model (discussed in Section 5). The full data model contains hundreds of types of resources that form deep tree-like structures [Managed Object, 2024]. Next, we consider how relationships between objects are modeled in DDO. Figure 1 already defines basic relationships, including 1:1 (dotted edges) and 1:m (solid edges).

**Many-to-many (m:m).** CLIENT and RESOURCEPOOL are related via a m:m relationship (the dashed edge) meaning that a client might use multiple resource pools and one resource pool can serve multiple clients. The table RSPOOL2CLIENT models this relation.

**Star.** A STAR pattern is a type of database schema composed of a single, central fact table surrounded by dimension tables. There are two groups of objects connected in a STAR patterns in our example. STAR A keeps track of retention marketing strategies for each client that can be either GIFT or/and BONUS. STAR B records clients' payments (PAYMENT). Payments' amounts are stored in the PAYAMOUNT table. Each amount can be exactly one of three types: TAX, SUPERCHARGE, and INCOME.

**Snowflake.** A SNOWFLAKE schema consists of one fact table connected to many dimension tables, which can be connected to other dimension tables through a many-to-one relationship. In DDO, database resource pools are modeled using the snowflake pattern. Each resource pool has configurations (CONFIG) and snapshots of the current usage (RUNTIME). CONFIG and RUNTIME have two children nodes each to define CPU and memory properties.

**Lookup.** A LOOKUP table is a table that contains descriptions and code values used by multiple tables, e.g., zip codes, country names. etc. In DDO, LOCATION is a lookup table that stores geo-location related data for quick access.

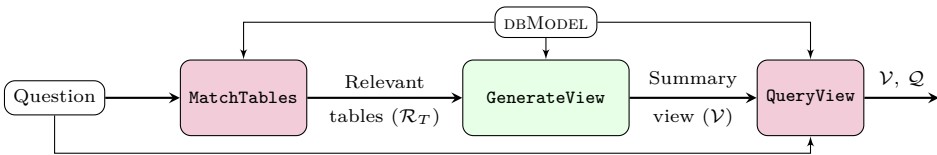

Figure 2: LUCY's high-level workflow. Red colored boxes indicate phases performed by LLMs, and a green colored box is a phase performed by an automated reasoner.

## 2.2 User questions

We consider three simple questions to DDO that are well formulated: outputs are explicitly specified, so no additional information is needed to answer them. We use GPT4 ('gpt-4-0125-preview'), and PROMPTB [Sequeda et al., 2023] for these questions. For each question, we present a ground truth answer and a GPT answer. Table 1 presents both questions (Q3 is presented in Appendix C.1).

Question Q1 is *'List customers who use datacenters with names starting with 'dev'. Output clients and datacenters names'*. The user asks for information that relates clients and datacenters. Consider GPT's answer. GPT misses the core logic of the database: *clients* and datacenter *resources* are related via a m:m relation (modeled with RSPOOL2CLIENT). GPT outputs clients and datacenters that share the same location, which is incorrect.

Question Q2 is *'List resource pool names with CPU overhead limit greater than runtime overall usage by 100'*. Here the user asks about resource pool properties. However, the GPT answer ignores the database's primary/foreign relations. It performs an `inner join` between RESOURCEPOOL, CCPU, and RCPU tables, using non-existent attributes RESOURCEPOOL.*config_id* and RESOURCEPOOL.*runtime_id*, which is clearly incorrect.

In summary, these examples demonstrated that LLMs struggle to handle complex relationships between objects.

## 3 Framework design

In this section, we present our framework LUCY. Figure 2 illustrates the workflow diagram, and Algorithm 1 shows the main steps of the workflow. There are two inputs to the framework. The first input is a user question $Q$. The second input is DBMODEL, which is a description of the database schema that we discuss in the next section (Section 3.1). The workflow consists of three sequential subtasks: `MatchTables`, `GenerateView`, and `QueryView`. `MatchTables` identifies the relevant tables and their attributes related to the user question (Section 3.2). `GenerateView` finds a combined view of relevant tables taking into account database constraints (Section 3.3). The third phase, `QueryView`, takes $\mathcal{V}$ and the user question $Q$ and produces an SQL query $\mathcal{Q}$ for $\mathcal{V}$ (Section 3.4). To simplify notations, we assume that DBMODEL is a global variable in Algorithm 1.

### 3.1 Database model (dbModel)

We start with DBMODEL, or DBM for short. DBM is a data structure that contains aggregated information about the database, maintained as a JSON structure. DBM should be constructed once for a database as the structure of the database is relatively stable. DBM can always be extended if the database requires modifications. Here are the two main blocks of DBM:

*Database schema.* The schema is written using the SQL Data Definition Language (CREATE TABLE statements). It includes table names, names and types of columns in each table, and database constraints such as primary and foreign keys. It can also contain optional user comments associated with each table and column. We refer to tables and constraints as DBM.tables and DBM.constraints, respectively. We extract this information in the form of JSON. Appendix D.1.1–D.1.2 shows examples of these structures.

153 *Patterns summary.* The user can optionally list higher-level design patterns that are not
154 captured by the schema explicitly. This information can help to improve the accuracy of the
155 algorithm. We support m:m, STAR, SNOWFLAKE, and LOOKUP patterns, but the model is
156 extendable to support other patterns. The user identifies these patterns manually, based on
157 the logic of the target domain. In the future, we envision that the process can be partially
158 automated. Appendix D.1.3 shows the JSON format used to specify pattern structures.

159 **Formal notations.** We introduce formal notations. DBM.tables contains a list of tables $t_i$,
160 $i \in [1, m]$ where $m$ is the number of tables. DBM.constraints contains a set of pairs $(t_i, t_j)$
161 such that $t_i$ and $t_j$ are related via 1:1, 1:m or m:1 relation. We denote DBM.m:m as a
162 set of triplets $(t_i, t_j, t_k)$, where a join table $t_k$ models a m:m relation between tables $t_i$
163 and $t_j$. Note that $(t_i, t_k)$ and $(t_j, t_k)$ must be in DBM.constraints. Additionally, we denote
164 DBM.LOOKUP as the set of lookup tables. For example, in the DDO database, DBM.m:m =
165 {(CLIENT, RESOURCEPOOL, RSPOOL2CLIENT)} and DBM.LOOKUP = {LOCATION}. For a
166 tree-like pattern, like STAR or SNOWFLAKE, we distinguish between root table and inner
167 tables using two predicates, e.g., `star_root`$(t)$ returns TRUE if $t$ is the root table of a STAR
168 and `star_inner`$(t)$ returns TRUE if $t$ is an inner table (not root) of a STAR.

### 3.2 The `MatchTables` phase

170 The first phase, `MatchTables`, needs to find relevant tables and their attributes to the user
171 question. One approach to achieve that can be to provide the schema and a question to an
172 LLM and ask for this information. However, one of the distinguishing features of real-world
173 databases is their large number of tables and attributes. Hence, feeding all of them along with
174 their descriptions to the prompt might not be feasible for many LLM models. Therefore, we
175 build an iterative procedure that takes advantage of database tree-like patterns. In general,
176 this procedure can be customized to best support the structure of a database.

---

**Algorithm 1** LUCY

**Require:** User question $Q$, database model DBMODEL
**Ensure:** Summary view $\mathcal{V}$, SQL query $\mathcal{Q}$
1: **Phase 1: MatchTables //LLM-based phase**
2: // get core tables (these are tables that are not inner tables in STAR or SNOWFLAKE)
3: core_tables = $\{t | t \in \text{DBM}.tables \wedge t \notin (\text{snowflake\_inner}(t) \vee \text{star\_inner}(t))\}$
4: // identify relevant core tables to the user query
5: _, $T$ = PROMPTA $(Q, \text{core\_tables}, \{\})$
6: $\mathcal{R}_T = \{\}$
7: **for** $t \in T$ **do**
8:     **if** $t \in \text{snowflake\_root}(t) \vee t \in \text{star\_root}(t)$ **then**
9:         // a breadth-first deepening to identify relevant tables and attributes inside a pattern rooted at $t$
10:         $\mathcal{R}_T = \mathcal{R}_T \cup \text{ITERATIVEPROMPTING}(Q, t)$
11:     **else**
12:         $\mathcal{R}'_T, \_ = \text{PROMPTA}(Q, \{\}, t.\text{attributes})$, $\mathcal{R}_T = \mathcal{R}_T \cup \mathcal{R}'_T$ // identify $t$'s relevant attributes
13: **Phase 2: GenerateView // constraint reasoner-based phase**
14: // formulate a constraint satisfaction problem
15: $S = \text{formulate\_csp}(\mathcal{R}_T)$
16: // solve $S$ to find a path in $G$ that satisfies constraints $(C_1)$–$(C_5)$
17: $\mathcal{P} = \text{solve\_csp}(S)$
18: // build a view $\mathcal{V}$ base on $\mathcal{P}$ by joining tables along the path $\mathcal{P}$.
19: $\mathcal{V} = \text{build\_view}(\mathcal{P})$
20: **Phase 3: QueryView //LLM-based phase**
21: $\mathcal{Q}= \text{PROMPTC} (Q, \mathcal{V})$
22: **return** $\mathcal{V}, \mathcal{Q}$

---

177 Algorithm 1 shows `MatchTables` in lines 2–12. First, the algorithm focuses on tables that are
178 not inner tables of any patterns. We refer to such tables as core tables (core_tables in line 3).
179 For example, Figure 3 shows core tables for DDO. Next, we ask LLM to find relevant tables
180 among these core tables using PROMPTA in line 5. (Appendix D.2.1 shows a PROMPTA with
181 a few examples.) As a result, we obtain a set of relevant core tables. We explore them one
182 by one in the loop in line 7. If it is a root table of a pattern, we perform a search inside the
183 corresponding pattern to find more relevant tables using a breadth-first deepening procedure,
184 ITERATIVEPROMPTING, in line 10 (Algorithm 2 shows ITERATIVEPROMPTING's pseudocode
185 in Appendix D.2). Otherwise, we use PROMPTA to obtain relevant attributes in line 12.

186 ***Example* 3.1.** *Consider questions Q1 and Q2 from Table 1. Figure 3 shows DDO's core*
187 *tables. For Q1, a LLM identifies relevant core tables: $T$ = {CLIENT, DATACENTER}*

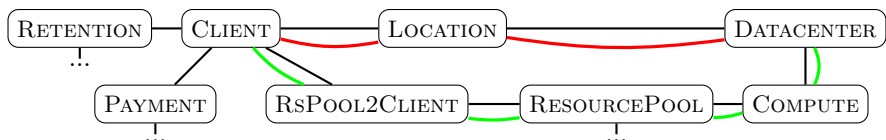

Figure 3: A part of the abstract schema graph $G$ for DDO that includes core tables.

*(line 5). Since none of these tables is a root of a SNOWFLAKE or a STAR, we prompt for relevant attributes for each table in line 12 to get $\mathcal{R}_T = \{$CLIENT.name, CLIENT.gender, DATACENTER.name$\}$. Now consider Q2. LLM identifies RESOURCEPOOL as a relevant table in line 5. As RESOURCEPOOL is the root table of SNOWFLAKE (see Figure 1), we begin to explore the pattern tree in a breadth-first order using ITERATIVEPROMPTING in line 10. RESOURCEPOOL has two child nodes, CONFIG and RUNTIME, and several attributes. We query the LLM and find that both CONFIG and RUNTIME are relevant as well as its attribute RESOURCEPOOL.name. Following the breadth-first search order, we consider CONFIG with two descendants cCPU and cMEMORY and discover cCPU is relevant (Example D.4 in Appendix shows a full version).*

### 3.3 The `GenerateView` phase

The `MatchTables` phase identifies a set of relevant tables and their attributes. Next, we construct a view table that combines relevant tables and attributes into a single table.

We build an abstract schema graph $G$ which provides a graph view of DBM, and define a CSP over this graph. For each table $t_i$ in DBM.tables, we introduce a node in $G$. We use the names $t_i$ to refer to the corresponding nodes. For each pair of tables $t_i$ and $t_j$, s.t. $(t_i, t_j) \in$ DBM.constraints, we introduce an edge that connects them. We denote $V$ the set of nodes in $G$ and $E$ its edges. Figure 3 illustrates a part of the graph (core tables) for DDO.

Algorithm 1 shows three main steps of this phase: build an abstract graph representation $G$ of the schema (line 15); formulate and solve CSP to obtain a path $\mathcal{P}$ (line 17); and perform joins along this path to obtain the designed view $\mathcal{V}$ (line 19). Next, we describe these steps.

**Problem Formulation.** Let $T = $ tables$(\mathcal{R}_T)$ be a set of relevant tables returned by `MatchTables`. We formulate the problem of finding a path $\mathcal{P}$ in $G$ that visits a set of nodes $T$ and satisfies a set of database constraints.

($C_1$) $\mathcal{P}$ must be a valid path in $G$. This ensures that we follow primary/foreign keys relationships, i.e., 1:1, 1:m, and build a valid view.

($C_2$) $\mathcal{P}$ visits all relevant tables $T$. This ensures combining all relevant tables to a view.

($C_3$) Consider $(t_i, t_j, t_k) \in$ DBM.m:m. If $t_i \in \mathcal{P}$ and $t_j \in \mathcal{P}$ then $t_k$ must occur in $\mathcal{P}$ once between $t_i$ and $t_j$. These constraints enforce m:m relationships.

($C_4$) If $t \in \mathcal{P}$ and $t \in$ DBM.lookup then $t$'s predecessor equals its successor in $\mathcal{P}$. This ensures that a lookup table serves as a look-up function for each table individually.

($C_5$) Cost function: we minimize the number of occurrences of tables outside of $T$ in $\mathcal{P}$. A shorter path that focuses on the tables in $T$ allows us to build more succinct views.

($C_1$)–($C_5$) are common constraints that we encounter in the benchmark sets. In general, the user can specify more constraints to capture the logical relationships of the modeled data. **Constraint satisfaction problem (CSP).** We define a CSP formulation $S$ of constraints ($C_1$)–($C_5$). We start with a basic formulation. Let $n$ be the maximum length of the path $\mathcal{P}$. For each node $t_i$ in $G$ and step $r$, where $r \in [1, n]$, we introduce a Boolean variable $b_i^r$. $b_i^r$ is true iff $t_i$ is the $r$th node in $\mathcal{P}$. We also introduce a sink-node Boolean variable $b_d^r$ for each layer to model paths that are shorter than $n$. $S$ contains the following logical constraints:

$$(C_5): \quad minimize \sum_{i, t_i \notin T} occ_i \tag{1}$$

$$\forall i. t_i \in V \qquad occ_i = b_i^1 + \ldots + b_i^n \tag{2}$$

$$(C_1): \quad \forall i. t_i \in V, r \in [1, n-1] \qquad b_i^r \Rightarrow (\vee_{j.(t_i, t_j) \in E} b_j^{r+1}) \vee b_d^{r+1} \tag{3}$$

$$(C_2): \quad \forall i. t_i \in T \qquad occ_i \geq 1 \tag{4}$$

$$(C_3): \quad \forall k.(t_i, t_j, t_k) \in \text{DBM.m:m} \qquad\qquad\qquad\qquad \text{occ}_k = 1 \quad (5)$$

$$(C_3): \quad \forall k.(t_i, t_j, t_k) \in \text{DBM.m:m}, r \in [2, n-1] \quad b_k^r \Rightarrow (b_i^{r-1} \wedge b_j^{r+1}) \vee (b_j^{r-1} \wedge b_i^{r+1}) \quad (6)$$

$$(C_4): \quad \forall i.t_i \in \text{DBM.lookup}, r \in [2, n-1] \qquad b_i^r \Rightarrow (b_j^{r-1} \Rightarrow b_j^{r+1}) \quad (7)$$

$$\forall r \in [1, n] \qquad\qquad\qquad\qquad b_1^r + \ldots + b_{|V|}^r = 1 \quad (8)$$

$$\forall r \in [1, n-1] \qquad\qquad\qquad\qquad b_d^r \Rightarrow b_d^{r+1} \quad (9)$$

Consider the encoding $S$. Equations 2 specify integer variables, $\text{occ}_i$, for $i \in [1, n]$, that count the occurrences of each table in the path. Equations 8 encode that only one node belongs to a path at each step. Equations 9 encode that if the path visits the sink node, then it must stay there. Other equations encode constraints $(C_1)$–$(C_5)$. By construction, Equations 1–9 generate a valid path in $G$ that satisfies the constraints $(C_1)$–$(C_5)$.

***Example* 3.2.** *For Q1, solving $S$ gives the green path between* DATACENTER *and* CLIENT *in Figure 3. $S$ rules out the red path as we enforce constraint $(C_4)$ and optimization $(C_5)$.*

**Improvements of CSP.** Our basic model $S$ can be improved to take advantage of STAR and SNOWFLAKE patterns. Namely, we can leverage the decomposition of $G$ and find a path $\mathcal{P}$ among core tables only. Then, for each core table in $\mathcal{P}$ that is a pattern root, and for each inner relevant table in this pattern, we build a path $\mathcal{P}'$ along the corresponding branch. For example, Figure 1 shows two paths from RESOURCEPOOL to cCPU (an orange path) and rCPU (a blue path). We use `left join` to combine tables along each such branch. Finally, we combine $\mathcal{P}$ and $\mathcal{P}$'s into a single view.

**Summary view.** Given a path $\mathcal{P}$ in a graph, we `join` tables along the path using their primary and foreign key relations. We keep the same set of attributes that `MatchTables` identified. An example of the $\mathcal{V}$ for Q1 that corresponds to the green path in Figure 3 is shown in the listing in Table 7 in Appendix D.3.1.

### 3.4 The `QueryView` phase.

`QueryView` takes the summary view $\mathcal{V}$ along with the user question, and prompts an LLM to obtain the final SQL using PROMPTC (line 21 in Algorithm 1). PROMPTC is defined in Appendix D.4.1. The listing in Table 7 shows an SQL $\mathcal{Q}$ to answer Q1 (Appendix D.3.1).

## 4 Discussion on strengths and limitations

**Strengths.** LUCY is designed based on the principle of separation of responsibilities between generative tasks and automated reasoning tasks: each step focuses on either an NLP-related subproblem or a constraint reasoning subproblem. This separation allows us to support a number of unique capabilities. First, LUCY shifts the burden of complex reasoning from LLMs to constraint solvers. Second, we support reasoning on complex relationships, like m:m, LOOKUP, STAR or SNOWFLAKE. Third, our framework is flexible and extensible as it is easy to incorporate domain-specific constraints as soon as they can be expressed by constraint modeling language. This assumes that the user has a data analytics role and understands the logic of the database. Such formal reasoning capability is important, as it is hard to control LLMs via prompts when non-trivial reasoning is required. Fourth, we can evaluate each phase and diagnose LUCY failure modes. For example, if `MatchTables` misses relevant tables, this indicates that we need to provide more information about the schema to an LLM. Fifth, based on our evaluation, LUCY can support complex queries that include multiple filtering operators and aggregators, e.g. average or sum. This capability follows from the `QueryView` phase as the final call to an LLM is performed on a single view table.

**Limitations.** The first limitation is that we cannot guarantee that the SQL query answers the user's question. Given the current state of the art, providing such guarantees is beyond the reach of any copilot method that takes natural language descriptions and outputs structured text, like code or SQL. However, our solution does guarantee that $\mathcal{V}$ satisfies database constraints, which is a step forward in this direction. Second, we do not support questions

that require `union` operators in the `GenerateView` phase. In fact, there are no benchmarks available that require the `union` operator to answer questions. Supporting `union` would require an extension of `MatchTables` and `GenerateView`. Third, we observed experimentally that LUCY struggles with certain types of queries that involve a particular interleaving ordering of filtering and aggregate operators or question-specific table dependencies, like a lookup table that has to be used multiple times to answer the user's question. We further discuss such questions in our experiments.

# 5   Experimental evaluation

In our experimental evaluation, we aim to answer the main questions:

- Is LUCY competitive with existing LLM-based approaches?
- Can we debug LUCY to gain insights about failure modes?
- Can LUCY handle complex questions?

**Setup.** We compare with the following zero-shot baselines: GPT4, NSQL, and CHAT2QUERY (C2Q for short). GPT4 and C2Q methods are the best zero-shot techniques according to the BIRD leadership board that are accessible for evaluation [Li et al., 2024b]. NSQL is the best open-source large foundation model designed specifically for the SQL generation task [Labs, 2023b]. CHAT2QUERY is closed-source but the authors kindly extended their API that we can run experiments with GPT4. We provide all benchmarks and frameworks' results in the supplementary materials. For GPT4 and LUCY, we use the 'gpt-4-0125-preview' API without fine-tuning. We use OR-Tools as a constraint solver [Perron and Didier, 2024] (Appendix E.1 provides full details of the experimental setup).

**Evaluation metrics.** We use the standard Execution Accuracy ($ex$) [Li et al., 2023]. In addition, we consider a relaxation of this metric. We noticed that frameworks often add additional attributes to the output as the exact format of the output is rarely specified. Hence, we extend $ex$ to $esx$ metrics that check if the output of a framework contains the ground truth outputs. To better understand performance characteristics and possible failure modes, we consider the coverage metric that captures whether a framework correctly identified a subset of relevant tables and attributes. Let $sql_G$ be the ground truth answer and $sql_F$ be a generated query. Then we assess the percentage of the ground truth content $slq_F$ captures:

$$cov_t = \frac{|\text{tables}(slq_F) \cap \text{tables}(slq_G)|}{|\text{tables}(slq_G)|} \quad cov_a = \frac{|\text{attributes}(slq_F) \cap \text{attributes}(slq_G)|}{|\text{attributes}(slq_G)|}, \quad (10)$$

where tables () and attributes () are functions that return a set of tables and attributes.

Table 2: The *ACME insurance* dataset.

|        | GPT4 | GPT4EX | C2Q  | NSQL | LUCY     | DW |
|--------|------|--------|------|------|----------|----|
| $cov_t$ | 0.44 | 0.47   | 0.82 | 0.31 | **0.95** | -  |
| $cov_a$ | 0.36 | 0.42   | 0.81 | 0.25 | **0.93** | -  |
| $ex$   | 9    | 13     | 16   | 2    | **30**   | 24 |
| $esx$  | 9    | 13     | 16   | 3    | **33**   | -  |

Table 3: The *Cloud Resources* dataset.

|        | GPT4 | GPT4EX | C2Q  | LUCY     |
|--------|------|--------|------|----------|
| $cov_t$ | 0.46 | 0.44   | 0.44 | **0.98** |
| $cov_a$ | 0.50 | 0.44   | 0.48 | **0.98** |
| $ex$   | 6    | 4      | 2    | **17**   |
| $esx$  | 9    | 5      | 2    | **18**   |

**ACME insurance.** We consider the *ACME insurance* dataset that was recently published [Sequeda et al., 2023]. The dataset represents an enterprise relational database schema in the insurance domain. The authors focused on a subset of 13 tables out of 200 tables and proposed a set of 45 challenging questions. We identified two STAR patterns in this database. The authors showed that their method (DW) solved 24 out of 45 problems using intermediate representation of a knowledge graph, while GPT4 solved only 8 problems. However, results are not publicly available, so we cannot perform coverage analysis and compute $esx$.

We reran the experiment on GPT4 with the same PROMPTB (Appendix C.1.1) and obtained similar results to those reported in [Sequeda et al., 2023]. In addition, we extended the schema with descriptions of table attributes from DBMODEL in the form of comments, which we called GPT4EX (See Appendix E.2 for examples). Table 2 shows our results. First, we observe that there is a strong correlation between coverage and accuracy metrics in the results. C2Q and LUCY show good coverage, meaning that they can correctly identify most of

the required tables and attributes. They also demonstrate better performance compared to other methods. Our framework shows very high coverage and solves about 30 of benchmarks according to the *ex* metric, which outperforms DW that solves 24 and other methods.

LUCY still cannot solve 13 benchmarks, which is surprising given high coverage. We performed a study to locate where LUCY fails on these benchmarks (See Appendix E.2.1 for all questions where LUCY was unsuccessful). In summary, the majority of failures come from under-specified output attributes or nonstandard aggregators, like specialized formulas to compute an average. In four cases, `MatchTables` missed a table, and in one case, `QueryView` missed the attribute to output. The most interesting mode of failure is when we need to perform multiple lookups on the same table. The reason for that is the `MatchTables` phase identifies only relevant tables but ignores possible relationships between them. Extending `MatchTables` to retrieve relationships between tables is interesting future work.

**BIRD datasets.** Next, we consider the state-of-the-art dataset BIRD [Li et al., 2023]. From the development set, we chose two datasets with complex relationships between objects: *financial* (106 instances) and *formula*1 (174 instances)[1]. The accuracy of CHAT2QUERY on the BIRD development set is $\sim 58\%$; however, its accuracy on *financial* and *formula*1 are much lower, $\sim 45\%$. We compare with results from GPT4'23 and C2Q available from [AlibabaResearch, 2020] and [TiDBCloud, 2020a], respectively. However, we reran these benchmarks with GPT4 and GPT4EX as the GPT4'23 results are nearly one year old. Table 5 and Table 4 show results on *financial* and *formula*1, respectively. LUCY and C2Q have higher coverage and good accuracy. LUCY shows the best results in most cases. Again, LUCY has very good coverage on *financial* but was able to solve only 68 out of 106 queries based on the *esx* metric. We manually performed an questions study on the failed questions. There are two major groups there that are interesting. First, LUCY has difficulty if there are multiple orderings, especially nested or ordering in different directions. Second, sometimes, `MatchTables` adds an additional table that is not needed to find the answer. The rest are either ambiguous questions or small mistakes like outputting a wrong attribute, i.e., *id* instead of *name*. See Appendix E.3.3 for examples of questions where LUCY was unsuccessful.

Table 4: The *formula*1 dataset.

|  | GPT4'23 | GPT4 | GPT4EX | C2Q | NSQL | LUCY |
|---|---|---|---|---|---|---|
| $cov_t$ | 0.86 | 0.78 | 0.77 | 0.88 | 0.52 | **0.93** |
| $cov_a$ | 0.84 | 0.75 | 0.75 | 0.81 | 0.50 | **0.94** |
| $ex$ | 54 | 67 | 65 | 80 | 9 | **83** |
| $esx$ | 66 | 80 | 79 | 93 | 10 | **103** |

Table 5: The *financial* dataset.

|  | GPT4'23 | GPT4 | GPT4EX | C2Q | NSQL | LUCY |
|---|---|---|---|---|---|---|
| $cov_t$ | 0.81 | 0.84 | 0.87 | 0.92 | 0.50 | **0.97** |
| $cov_a$ | 0.81 | 0.81 | 0.85 | 0.91 | 0.59 | **0.96** |
| $ex$ | 36 | 47 | 52 | **59** | 6 | 56 |
| $esx$ | 38 | 55 | 64 | 62 | 6 | **68** |

**Cloud resources.** Next, we propose a new benchmark based on the vSphere API data model [VMware, Inc., 2024]. We experimented with this publicly available data model of an industrial product, as it is well-documented and easily accessible via a web interface. It describes the state of the system as well as its configuration parameters. States are stored in a database and queried by customers to keep track of performance, maintenance, and data analysis. We extracted the descriptions of main objects in Managed Object [2024], including data centers, resource pools, hosts, and virtual machines and their properties, and built a database that captures these relationships using 52 tables. Overall, we have two STARs, five SNOWFLAKEs and two m:ms patterns. For each table and an attribute, we get descriptions from [Managed Object, 2024]. As these can be a lengthy description, we use GPT to shorten it to 15 words (see PROMPTD in Appendix E.3.2) . We generated data randomly using sqlfaker [Kohlegger, 2020]. We create 20 challenging questions for this benchmark.

Table 3 shows our results. NSQL cannot process this benchmark due to a limited context window. We again see that LUCY outperforms other models in both coverage and accuracy. C2Q failed on 6 questions with an error 'Unable to generate SQL for this database due to its extensive tables' and it often does not follow instructions on the output columns. In terms of failure mode, LUCY failed in the third phase as it hallucinated some attribute names when names are long, e.g., 'Resourcepoolruntimemory' instead of 'Resourcepoolruntimememory'.

---

[1] Recently, Wretblad et al. [2024b] provided a detailed analysis of the BIRD dataset and found a number of errors of various types. See Appendix E.3 for the discussion.

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

## A    Background

**Relational databases.**    Let $D_1, \ldots, D_n$ be a set of domains. A relation or table, $t$, is defined over subset of domains: $t(X_{i_0}, \ldots, X_{i_k}) \subseteq D_{i_0} \times \ldots \times D_{i_k}$, $X_{i_j} \subseteq D_{i_j}, j \in [0, k]$. In addition, $t$ defines a set of attributes (or columns) names $X_{i_0}, \ldots, X_{i_k}$. Projection is a unary operation on a set of attribute names $Y$, $Y \subseteq X$. The result of such projection is the set of tuples that is obtained when all tuples in $t$ are restricted to attributes $Y$. Inner join, or simply `join`, is a binary operator between two tables $t_1$ and $t_2$ over their common attributes $Y$ that returns a set of all combinations of tuples in $t_1$ and $t_2$ that are equal on $Y$. Left join, `left join`, is similar to the join but returns all rows of $t_1$ filling unmatched rows of $t_2$ with null values. A database can support a large set of constraints over tables. The two main constraint types are related to primary and foreign keys. A primary key is the smallest subset of attributes guaranteed to uniquely differentiate each tuple in a table. A foreign key is a subset of attributes $Y$ in a table $t_1$ that corresponds with (usually) a primary key of another table $t_2$, with the property that the projection of $t_1$ on $Y$ is a subset of the projection of $t_2$ on $Y$ [Beaulieu, 2009].

**Design patterns.**    A database typically represents entities and their interactions in real-world processes, e.g., the financial management of a company. To effectively model these complex entities, several design patterns have been developed [IBM, Inc., 2021, Silverston et al., 1997]. A many-to-one pattern (`m:1`) specifies a relationship when any number of attributes from one table is associated with unique attributes of the same or another table, typically enforced by foreign key and primary key relationships. A many-to-many relationship (`m:m`) occurs when any number of attributes from one table is associated with any number of attributes from the same or another table. It is typically modeled with an auxiliary join table that refers to the primary keys of the tables in the relationship. The LOOKUP table is a table that contains descriptions and code values used by multiple tables, e.g., zip codes, country names. etc. A STAR pattern is a type of relational database schema composed of a single, central fact table surrounded by dimension tables. A SNOWFLAKE schema consists of one fact table connected to many dimension tables, which can be connected to other dimension tables through a many-to-one relationship.

**Constraint satisfaction.**    A constraint satisfaction problem (CSP) consists of a set of variables, each with a finite domain of values, and a set of constraints specifying allowed combinations of values for subsets of variables [Rossi et al., 2006]. A solution is an assignment of values to the variables satisfying the constraints. In the constraint optimization problem, we are looking for a solution that optimizes a given cost function. Constraint solvers typically explore partial assignments enforcing a local consistency property using either specialized or general-purpose propagation algorithms and employ conflict-driven learning to store information from failures as the search proceeds. We used OR-Tools CP-SAT solver [Perron and Didier, 2024] in our experiments.

## B    Related work

We focus on the zero-shot text-to-SQL problem, which has received significant attention in the last few years. Liu et al. [2023] performed a comprehensive evaluation of ChatGPT on the Spider dataset and demonstrated that it shows good performance. In [Dong et al., 2023], a new framework based on the GPT model was proposed, involving several techniques for promoting and post-processing the output to get more consistent results. Chang and Fosler-Lussier [2023] proposed several techniques to improve the performance of ChatGPT. [TiDBCloud, 2020b] represents the most recent zero-shot method. According to the API documentation [TiDBCloud, 2020b], the authors construct a data summary object that contains 'AI exploration information of the given database.' This method performs very well on the BIRD dataset. However, it relies on LLMs to reason about database relationships. Sequeda et al. [2023] performed an interesting investigation of the performance of LLMs on large industrial databases. They identified that GPT does not perform well when it needs to reason about complex relationships. The authors proposed a two-step approach to tackle this problem. As a knowledge graph is available for these benchmarks, the authors proposed

| |
|---|
| *Q3: What are the total tax payment, which is the sum of Tax and Supercharge?* |

```
/*GPT4 generated SQL*/: select sum(TAX.amount + SUPERCHARGE.amount)
from TAX
join PAYAMOUNT on PAYAMOUNT.id = TAX.payamt_id
join SUPERCHARGE on PAYAMOUNT.id = SUPERCHARGE.payamt_id
```

```
/*Correct SQL*/:
select sum(ifnull(PAYTAX.amount, 0)) + sum(ifnull(PAYSUPERCHARGE.amount,0))
from PAYMENT
join PAYAMOUNT as PAYTAX on PAYMENT.id = PAYTAX.pay_id
left join TAX on PAYTAX.id = TAX.payamt_id
join PAYAMOUNT as PAYSUPERCHARGE on PAYMENT.id = PAYSUPERCHARGE.pay_id
left join SUPERCHARGE on PAYSUPERCHARGE.id = SUPERCHARGE.payamt_id
```

Table 6: A user's question Q3. Incorrect parts of the GPT answer are highlighted in red.

using the knowledge graph as an intermediate representation. Namely, the user's question is answered using the *KG* structure with SPARQL, and this answer is automatically translated to SQL using a given mapping from ontology to SQL (R2RML). However, while reasoning on a knowledge graph can be easier for LLMs, it is still challenging to take all complex relationships into account.

## C   Motivation (additional materials)

### C.1   User's questions

The third question Q3, *'What are the total tax payments, which is the sum of Tax and Supercharge?'*, asks about the total amount of taxes paid from all payments (Table 6). There are a few issues with the GPT answer. First, it outputs all payment amounts that are both tax and supercharge. We reminded that that each payment amount can be of one type, so the result will be empty. Second, it hallucinates as there are no *amount* columns in the TAX or SUPERCHARGE tables.

### C.1.1   Definition of promptB

> **Inputs:**   DB_SCHEMA, Question
> **promptB**   Given the database described by the following DDL: <DB_SCHEMA>.
> Write a SQL query that answers the following question. Do not explain the query.
> Return just the query, so it can be run verbatim from your response. Here's the
> question: <Question>. [Sequeda et al., 2023]
> **Returns**   : SQL

## D   Framework design (additional materials)

### D.1   Database model (dbModel)

### D.1.1   Example of a table from dbm.tables

Here is a JSON structure for the Client table from the *financial* dataset [Li et al., 2023]. It contains the table name, primary keys, attributes, their types, and descriptions. This information is available in the dataset. The description of the table is generated by GPT4 using the prompt PROMPTD.

```
"Client": {
    "type": "ManagedObject",
    "primary": [
        "client_id"
    ],
```

```
545    "path": "<path -to >/ Client.json",
546    "path_to_types": ""<path -to >/ Client_types.json"
548 }
547
```

Here is the JSON structure for Client.json:

```
550
5551 {
552     "NameField": "Client",
553     "DescriptionField": "Focuses on client information,
554        encompassing unique client identifiers, gender, birth
555        dates, and the location of the branch with which they
556        are associated.",
557    "client_id": "the unique number",
558    "gender": "  Description: 'F: female; M: male '",
559    "birth_date": "birth date",
560    "district_id": "location of branch"
561 }
562
```

Here is the JSON structure for Client_types.json:

```
564
5651 {
562     "NameField": {
563         "type": "varchar (100)",
564         "default": "DEFAULT NULL"
565     },
576     "DescriptionField": {
577         "type": "varchar (5000)",
578         "default": "DEFAULT NULL"
579     },
570     "client_id": {
571         "type": "bigint",
572         "default": "NOT NULL"
573     },
574     "gender": {
575         "type": "varchar (46)",
576         "default": "NOT NULL"
577         },
578     "birth_date": {
579         "type": "date",
580         "default": "NOT NULL"
581     },
582     "district_id": {
583         "type": "bigint",
584         "default": "NOT NULL"
585     }
586 }
591
```

#### D.1.2 Example of a m:1 relation from dbm.constraints

Here is the JSON structure for the Client and District relation from the *financial* dataset [Li et al., 2023].

```
595
5961   "Client, District": {
592         "type": "Relationships",
598         "sqlrelation": "M:1",
5994         "foreign_relation": {
605             "FOREIGN": [
616                 "district_id"
627             ],
```

```
608                    "foreign_relation_ref_table": "District",
609                    "foreign_relation_ref_table_keys": [
610                        "district_id"
611                    ]
612                }
613        }
609
```

### D.1.3    Example of a `m:m` pattern from dbm.patterns

Here is the JSON structure for the Account and District `m:m` relation from the *financial* dataset [Li et al., 2023].

```
{
    "Account, Client": {
            "type": "Relationships",
            "description": "",
            "sqlrelation": "M:M",
            "m2m_relation": {
                    "m2m_middle_table": "Disp",
                    "m2m_side_tables": [
                            "Client",
                            "Account"
                    ],
                    "m2m_relation_one": [
                            "Disp",
                            "Client"
                    ],
                    "m2m_relation_two": [
                            "Disp",
                            "Account"
                    ]
            }
        }
}
```

Here is the JSON structure for the SNOWFLAKE pattern rooted ta RESOURCEPOOL (*Cloud Resources* dataset).

```
{
        "NameField": "ResourcePool",
        "config": {
                "cpualloc",
                "memalloc"
        },
        "runtime": {
                "cpu",
                "memory"
        }
}
```

## D.2    MatchTables

### D.2.1    promptA

PROMPTA requires three inputs: a user question, a set of tables (can be empty), and a set of attributes for a given table $t$ (can be empty).

> **Inputs:** Question, Tables, Attributes
> **promptA:** Here is a json schema. Please treat json schema objects as a description of tables in a database <JSON(Tables, Attributes)>. The user has a query to answer <Question>. What are all relevant json elements to a user query from the list [<list of json elements>]? Output is a list of elements, [element, element, element,...]. Do not explain.
> **Returns:** We post-process the output to extract a set of tables and their attributes ($\mathcal{R}_T$) and relevant tables $T$

In the prompt, we provide description of tables and attributes from DBM. We show a few examples of <JSON(Tables, Attributes)> and the corresponding <list of json elements>.

***Example* D.1.** *Here is an example of a JSON(Tables, {}) used in line 5 in Algorithm 1 for financial dataset. The goal is to determine relevant core tables.*

```
{
    "Account": "Manages financial accounts, tracking each
        account's unique identification, the location of the
        associated bank branch, the frequency of account
        servicing, and the account's creation date. It
        categorizes the servicing frequency with options like
        monthly, weekly, and post-transaction issuances
        Properties of Account: account_id, district_id,
        frequency, date. ",
    "Card": "Manages of credit cards, incorporating unique
        identifiers for each card and the related
        dispositions. It also categorizes credit cards into
        various classes, such as junior, standard, and
        high-level, reflecting their tier and associated
        benefits. Properties of Card: card_id, disp_id, type,
        issued. ",
    "Client": "Focuses on client information, encompassing
        unique client identifiers, gender, birth dates, and
        the location of the branch with which they are
        associated. Properties of Client: client_id, gender,
        birth_date, district_id. ",
    "Disp": "Manage dispositions in financial accounts. It
        contains a unique identifier for each record, links
        each disposition to specific clients and accounts,
        and categorizes the nature of each disposition into
        types like 'OWNER', 'USER', or 'DISPONENT'.
        Properties of Disp: disp_id, client_id, account_id,
        type. ",
    "District": "Provides a detailed overview of
        district-level data, essential for regional analysis
        and decision-making. It includes a unique identifier
        for each district, along with the district's name and
        its broader region. The table delves into
        demographic, economic data and  economic indicators,
        records crime statistics. Properties of District:
        district_id, A2, A3, A4, A5, A6, A7, A8, A9, A10,
        A11, A12, A13, A14, A15, A16. ",
    "Loan": "Manages loan-related data, offering insights
        into each loan's unique identifier, associated
        account details, approval dates, amounts, durations,
        and monthly payments. Properties of Loan: loan_id,
        account_id, date, amount, duration, payments, status.
        ",
```

```
   "Order_": "Manages payment orders, detailing unique
      identifiers for each order, linked account numbers,
      and recipient bank details. It captures the bank and
      account number, the debited amount for each order and
      categorizes the purpose of each payment. Properties
      of Order_: order_id, account_id, bank_to, account_to,
      amount, k_symbol. ",
   "Trans": "Includes transaction management, encompassing
      details such as transaction identifiers, associated
      account numbers, and dates of transactions,
      categorizes transactions, covering a range of
      activities from insurance payments and statement fees
      to interest credits, sanctions for negative balances,
      household payments, pension disbursements, and loan
      payments; and details about the transaction partner's
      bank, identified by a unique two-letter code, and
      their account number. Properties of Trans: trans_id,
      account_id, date, type, operation, amount, balance,
      k_symbol, bank, account. "
}
```

*The list of JSON elements is as follows*

```
['Account', 'Card', 'Client', 'Disp', 'District', 'Loan',
   'Order_', 'Trans']
```

**Example D.2.** *Here is an example of a JSON({}, Attributes) used in line 11 in Algorithm 1 for the table District to determine relevant attributes (from the financial dataset).*

```
{
   "DescriptionField": "Provides a detailed overview of
      district-level data, essential for regional analysis
      and decision-making. It includes a unique identifier
      for each district, along with the district's name and
      its broader region. The table delves into
      demographic, economic data and  economic indicators,
      records crime statistics.",
   "district_id": "location of branch",
   "A2": "district_name",
   "A3": "region",
   "A4": "",
   "A5": "municipality < district < region",
   "A6": "municipality < district < region",
   "A7": "municipality < district < region",
   "A8": "municipality < district < region",
   "A9": "  Description: not useful",
   "A10": "ratio of urban inhabitants",
   "A11": "average salary",
   "A12": "unemployment rate 1995",
   "A13": "unemployment rate 1996",
   "A14": "no. of entrepreneurs per 1000 inhabitants",
   "A15": "no. of committed crimes 1995",
   "A16": "no. of committed crimes 1996"
}
```

*The list of json elements is as follows*

```
[district_id, A2, A3, A4, A5, A6, A7, A8, A9, A10, A11, A12,
   A13, A14, A15, A16]
```

Moreover, if a table is a root table of a pattern, we provide inner tables and their attribute names so that an LLM can determine the relevance of Snowflake to the user question.

**Example D.3.** *Here is an example of the Snowflake summary rooted at ResourcePool from the Cloud Resources benchmark.*

```
"ResourcePool": "Resource pools manage VM resources within a
    hierarchy, ensuring efficient allocation through
    configurable settings and states. Properties of
    ResourcePool: namespace, name, owner, summary, config,
    config, config.changeVersion, config.entity,
    config.lastModified, config.scaleDescendantsShares,
    config.cpualloc, config.cpualloc,
    config.cpualloc.expandableReservation,
    config.cpualloc.limit_, config.cpualloc.overheadLimit,
    config.cpualloc.reservation, config.cpualloc.shares,
    config.cpualloc, config.memalloc, config.memalloc,
    config.memalloc.expandableReservation,
    config.memalloc.limit_, config.memalloc.overheadLimit,
    config.memalloc.reservation, config.memalloc.shares,
    config.memalloc, config, runtime, runtime,
    runtime.overallStatus, runtime.sharesScalable,
    runtime.cpu, runtime.cpu, runtime.cpu.maxUsage,
    runtime.cpu.overallUsage, runtime.cpu.reservationUsed,
    runtime.cpu.reservationUsedForVm,
    runtime.cpu.unreservedForPool,
    runtime.cpu.unreservedForVm, runtime.cpu, runtime.memory,
    runtime.memory, runtime.memory.maxUsage,
    runtime.memory.overallUsage,
    runtime.memory.reservationUsed,
    runtime.memory.reservationUsedForVm,
    runtime.memory.unreservedForPool,
    runtime.memory.unreservedForVm, runtime.memory, runtime,
    ResourcePool_id. "
```

### D.2.2 Description of the IterativePrompting algorithm.

---
**Algorithm 2** IterativePrompting
---
**Require:** $Q, t$
**Ensure:** Relevant tables and attributes in a tree-like pattern rooted at $t$
1: $stack\_tables = [t]$
2: $\mathcal{R}_T = \{\}$
3: **while** $stack\_tables$ **do**
4:      $r = stack\_tables.pop()$
5:      // check if $r$ is a leaf in a tree-like pattern
6:      **if** leaf($r$) **then**
7:          $\mathcal{R}'_T, \_ = $ PromptA($Q, \{\}, r$.attributes)
8:      **else**
9:          // find children of $r$ in a tree-like pattern
10:          children\_tables $= \{t | t \in $ DBM.$tables \cap $ children($r$)$\}$ // children(r) returns descendants of $r$ in the pattern.
11:          $\mathcal{R}'_T, T = $ PromptA($Q,$ children\_tables, $r$)
12:          $stack\_tables.push(T)$
13:      $\mathcal{R}_T = \mathcal{R}_T \cup \mathcal{R}'_T$
---

**Example D.4** (Full version of Example 3.1 for the question Q2)**.** *Consider Q2 from Table 1. Figure 3 shows* DDO*'s core tables. LLM identifies* ResourcePool *as a relevant table in line 5, along with its attribute* ResourcePool.name*. Since* ResourcePool *is the root table of a* Snowflake *pattern, we begin to explore the pattern tree in a breadth-first order using* IterativePrompting *in line 10. See Figure 1 for the structure of the the the* Snowflake *pattern.* ResourcePool *has two child nodes,* Config *and* Runtime*, and several attributes. We then query an LLM and find that both* Config *and* Runtime *are relevant as well its attribute* ResourcePool.name*. Following the breadth-first search order,*

```
/*--Summary view 𝒱 --*/
create view 𝒱 as select
CLIENT.id as CLIENT_id,
CLIENT.name as CLIENT_name,
CLIENT.gender as CLIENT_gender,
DATACENTER.name as DATACENTER_name,
DATACENTER.id as DATACENTER_id
from DATACENTER
join COMPUTE on DATACENTER.id = COMPUTE.dc_id
join RESOURCEPOOL on COMPUTE.id = RESOURCEPOOL.compute_id
join RSPOOL2CLIENT on RESOURCEPOOL.id = RSPOOL2CLIENT.rspool_id
join CLIENT on CLIENT.id = RSPOOL2CLIENT.client_id

/*--Final query 𝒬--*/
select CLIENT_name,
DATACENTER_name
from 𝒱 where DATACENTER_id > 1;
```

Table 7: `GenerateView` and `QueryView` results for *Q*1.

*we next consider* CONFIG *which has descendants* cCPU *and* cMEMORY *and a few attributes.*
*We discover that only one of them,* cCPU, *is relevant. We then move to the next table in*
*order,* RUNTIME. *It has two descendants* rCPU *and* rMEMORY *and a few attributes. We*
*discover that only one of them,* rCPU, *is relevant. Next, we identify relevant attributes of*
cCPU *in line 7 (Algorithm 2) and find that* cCPU.*overheadlimit is relevant to the user*
*query. Finally, we identify relevant attributes of* rCPU *in line 7 in (Algorithm 2) and find*
*that* rCPU.*overallusage is relevant to the user query.*

### D.3 The `GenerateView` phase

#### D.3.1 Summary view.

Consider again the question Q1 from Example 3.1. The view 𝒱 that corresponds to the
green path in Figure 3 is shown in the listing in Table 7. We keep the same set of attributes
that `MatchTables` identified. In addition, we also perform renaming of all attributes, as we
can control the length of the aliases (in case they are too long). For example, CLIENT.*name*
gets an alias CLIENT_*name*, CLIENT.*gender* gets CLIENT_*gender*, so on.

### D.4 The `QueryView` phase

#### D.4.1 promptC

Here is PROMPTC that we use in the final phase `QueryView` (Algorithm 1, line 21). The
function name() returns name of the view 𝒱.

---

**Inputs:** Question, 𝒱
**promptC**  I created a view table <name(𝒱)> with all relevant information. Here
is a view <𝒱 >. Please write MySQL query to name(𝒱) view to answer the following
question: <Question>. Use only name(𝒱) columns in the query. Absolutely NO
columns renaming. Absolutely NO HAVING operators. Absolutely NO COUNT(*).
Output query that I can run via python interface. Output '"'sql...'. Do not explain.
**Returns:** SQL

---

We used a few assertive statements that we discuss next. 'Absolutely NO column renaming'
means that we want to use aliases in the view table to form a valid SQL query. The statement
'Absolutely NO HAVING operators.' reflects our observation that GPT4 cannot generate
valid SQL when using HAVING in combination with GROUP BY. It is a subject of future
research to deal with MySQL constraints, so we encourage `QueryView` to avoid this operator.

832 Finally, we discourage the use of COUNT(*), 'Absolutely NO COUNT(*)', to ensure that
833 GPT4 focuses on counting the entities specified in the user's question.

834 We noticed that better results are obtained if we provide a description of tables that are used
835 to generate this view together with their relevant attributes. Here is an extended version of
836 PROMPTC where we provide relevant tables and their attributes that are used to obtain the
837 $\mathcal{V}$. We also provide an evidence if available.

---

**Inputs:** Question, $\mathcal{V}$, DB_SCHEMA
**promptC' (with evidence and a part of the schema)** Here is a SQL schema
for in MySQL: <DB_SCHEMA> I created a view table <name($\mathcal{V}$)> with all relevant
information. Here is a view $<\mathcal{V}>$. Please write MySQL query to name($\mathcal{V}$) view
to answer the following question: <Question>. Additional knowledge to answer:
<Evidence> Use only name($\mathcal{V}$) columns in the query. Absolutely NO columns
renaming. Absolutely NO HAVING operators. Absolutely NO COUNT(*). Output
query that I can run via python interface. Output '"'sql...'. Do not explain.
**Returns:** SQL

---

838

## E   Experimental evaluation (additional materials)

### E.1   Setup

841 We run experiments on a laptop Intel(r) Core 2.40Hz and 32GB of memory. For NSQL we use
842 the largest model with 7B parameters (NumbersStation/nsql-llama-2-7B [Labs, 2023a]). For
843 GPT4 and LUCY, we use the 'gpt-4-0125-preview' model as a LLM and set the temperature
844 to 0.2 . We do not fine-tune a LLM. We require 20 answers from GPT4 for each question. If
845 the number of correct answers is more than 5, then we count that benchmark as solved.

846 In the case of LUCY, we require 5 answers for each GPT call for the `MatchTables` phase.
847 We sort tables based on the number of occurrences in these answers and take at most 8
848 candidates among relevant tables from each PROMPTA output. Similarly to GPT4, we require
849 20 answers from `QueryView` and decide on the success as described above. We use ORTools
850 as a constraint solver [Perron and Didier, 2024].

851 We support MySQL as a relational database. However, BIRD uses SQLite. We automatically,
852 converted queries from sqlite to MySQL.

853 We provide all benchmarks and their results in the supplementary materials.

### E.2   *ACME insurance*

855 **Note on the database.** There are a few issues with broken relational constraints due to
856 missing tables, as reported [datadotworld, Inc., 2024], which we fixed by adding the missing
857 tables from the original database.

858 **Extended schema examples.** Example of tables extended with comments that describe
859 each attribute for the *ACME insurance* benchmark.

```
CREATE TABLE Claim_Amount
(
        Claim_Amount_Identifier bigint  NOT NULL COMMENT Claim Amount
            Identifier is the unique identifier of the financial
            amount reserved, paid, or collected in connection with a
            claim. The money being paid or collected for settling a
            claim and paying the claimants, reinsurers, other
            insurers, and other interested parties. Claim amounts are
            classified by various attributes.,
        Claim_Identifier     int  NOT NULL COMMENT Claim Identifier
            is the unique identifier for a Claim.,
        Claim_Offer_Identifier int  NULL COMMENT Claim Offer
            Identifier is the unique identifier for a Claim Offer.,
```

```
874    Amount_Type_Code        varchar(20)   NULL COMMENT Amount Type
875        Code defines the category to which a monetary amount will
876        be applied. Example:  premium, commission, tax,
877        surcharge.,
878    Event_Date              datetime   NULL COMMENT Event Date is the
879        date on which a transaction or insurance-related
880        happening takes place.,
881    Claim_Amount            decimal(15,2)   NULL COMMENT The money
882        being paid or collected for settling a claim and paying
883        the claimants, reinsurers, other insurers, and other
884        interested parties. Claim amounts are classified by
885        various attributes.,
886    Insurance_Type_Code  char(1)  NULL COMMENT  Insurance Type
887        Code represents the category under which risk is assumed.
888         Examples: Direct for policies directly issued by a
889        company; Assumed for risks assumed from another company;
890        Ceded for portions of risk ceded to another insurer.,
891     PRIMARY KEY (Claim_Amount_Identifier ASC),
892     FOREIGN KEY (Claim_Offer_Identifier) REFERENCES
893        Claim_Offer(Claim_Offer_Identifier),
894  FOREIGN KEY (Claim_Identifier) REFERENCES Claim(Claim_Identifier)
895 )
896

897
898
899 CREATE TABLE Claim_Reserve
900 (
901    Claim_Amount_Identifier bigint  NOT NULL COMMENT Claim Amount
902        Identifier is the unique identifier of the financial
903        amount reserved, paid, or collected in connection with a
904        claim. The amount of expected loss over the life of the
905        Claim.,
906     PRIMARY KEY (Claim_Amount_Identifier ASC),
907     FOREIGN KEY (Claim_Amount_Identifier) REFERENCES
908        Claim_Amount(Claim_Amount_Identifier)
909 )
910
```

### E.2.1  Challenging questions

In this section, we present 13 questions that LUCY found challenging to answer and identify reasons for these failures.

> **Question1:**  What are the loss payment, loss reserve, expense payment, expense reserve amount by claim number and corresponding policy number, policy holder, premium amount paid, the catastrophe it had, and the agent who sold it?
> **Reason:**  Multiple lookups. "policy holder" and "agent" require a look up to the same table Agreement_Party_Role.

> **Question2:**  What are the total loss, which is the sum of loss payment, loss reserve, expense payment, expense reserve amount by claim number and corresponding policy number, policy holder and premium amount paid?
> **Reason:**  Phase 1 issue. Phase 1 misses the relevant table Agreement_Party_Role.

> **Question3:**  What is the total amount of premiums that a policy holder has paid?
> **Reason:**  Phase 3 issue. Phase 3 makes a mistake in the group by clause.

**Question4:** What are the total loss, which is the sum of loss payment, loss reserve, expense payment, expense reserve amount by catastrophe and policy number?
**Reason:** Ambiguous question. By "by catastrophe", the user means to output Catastrophe's attribute Name. However, Phase 1 identifies Catastrophe's attribute Identifier as relevant instead of Name.

**Question5:** What is the average policy size which is the the total amount of premium divided by the number of policies?
**Reason:** Ambiguous question. The definition of average is not standard, as the same policy can have multiple *amount* values.

**Question6:** What are the loss payment, loss reserve, expense payment, expense reserve amount by claim number and corresponding policy number, policy holder, premium amount paid and the agent who sold it?
**Reason:** Multiple lookups.

**Question7:** Return agents and the policy they have sold that have had a claim and the corresponding catastrophe it had.
**Reason:** Ambiguous question. The output includes Company_Claim_Number, although this information is not specified in the question.

**Question8:** What is the loss ratio of each policy and agent who sold it by policy number and agent id?
**Reason:** Ambiguous question. "the loss ratio" is a complex formula here, making it hard to guess without its proper specification.

**Question9:** What are all the premiums that have been paid by policy holders?
**Reason:** Ambiguous question. Policy.Policy_Number and Party_Identifier should be included in the output. But they are not specified in the question.

**Question10:** What are the loss payment, loss reserve, expense payment, expense reserve amount by claim number and corresponding policy number, policy holder and premium amount paid?
**Reason:** Phase 1 issue. Phase 1 misses the relevant table Agreement_Party_Role.

**Question11:** What is the loss ratio, number of claims, total loss by policy number and premium where total loss is the sum of loss payment, loss reserve, expense payment, expense reserve amount and loss ratio is total loss divided by premium?
**Reason:** Phase 1 issue. Phase 1 misses the relevant table Policy.

**Question12:** What are the total loss, which is the sum of loss payment, loss reserve, expense payment, expense reserve amount by claim number, catastrophe and corresponding policy number?
**Reason:** Phase 1 issue. Phase 1 misses the relevant table Catastrophe.

> **Question13:** What is the total amount of premiums that a policy holder has paid by policy number?
> **Reason:** Ambiguous question. Party_Identifier is included in the output. But it is not specified in the question.

## E.3 BIRD datasets

### E.3.1 Additional notes on the dataset.

**Note on dbModel.** We used attribute descriptions available in BIRD in DBMODEL. We also build table descriptions in the following way. We provided the description from BIRD to an LLM to generate a short summary description using PROMPTD defined in Section E.3.2.

**Note on datasets.** It has been shown that there are a number of incorrect ground truth SQLs in BIRD datasets [Hui, 2024, Wretblad et al., 2024b]. For example, Wretblad et al. [2024b] found that 72 out of 106 benchmark questions in *financial* have errors of various types. Most of the issues have been reported to the authors from multiple sources, and we also reported additional problems via private communication. The authors acknowledge these issues and are working on them. To provide an example we reported from *formula*1:

- Question: 'Where can the introduction of the races held on Circuit de Barcelona-Catalunya be found?'
- Ground truth SQL: `select distinct` circuits.url `FROM` circuits `inner join` races `ON` races.circuitId = circuits.circuitId `where` circuits.name = 'Circuit de Barcelona-Catalunya'.
- The issue is that `select` should be on race.url rather than circuits.url as the user requests information about the race, not the circuit.

On top of that, there are *logical inconsistencies* in ground truth answers for the *financial* dataset. Often, users ask for information about clients' accounts. Client and account tables have a m:m relationship modeled using an additional table disp. At the same time, they are both related to a lookup table district. Unfortunately, many ground truth SQL statements perform a join between clients and accounts via the district table, which is incorrect. Let's consider an example.

- Question: 'Please provide the IDs of the 3 female clients with the largest loans.'
- Ground truth SQL: `select` T1.client_id `FROM` client `AS` T1 `inner join` account `AS` T2 `ON` T1.district_id = T2.district_id `inner join` loan `AS` T3 `ON` T2.account_id = T3.account_id `where` T1.gender = 'F' `order by` T3.amount DESC LIMIT 3
- The issue is that the answer relates clients and accounts that have the same district_id. However, this does not mean that the client is related to the account.

As the authors are working on corrections, we analyzed the reported issues and manually corrected the ground truth. We only adjusted the SQL ground truth values to match the user questions; we did not alter the questions or evidences. We provide the corrected benchmarks in the supplementary material. Specifically, we corrected the *financial* and *formula*1 benchmarks and used the correct answers to evaluate all methods. Interestingly, the performance of all frameworks improved by a few percentage points when we fixed these ground truth SQL statements.

### E.3.2 promptD

Here is PROMPTD that we use to generate tables summaries for *financial* and *formula*1 datasets.

> **promptD** Give me a very brief description of the <NAME> table.

> **Example for the Loan table (the** *financial* **database)**
> **Prompt:** Give me a very brief description of the Loan table.
> original_column_name,column_name,column_description,data_format,
> value_description
> loan_id„the id number identifying the loan data,integer,
> account_id„the id number identifying the account,integer,
> date„the date when the loan is approved,date,
> amount„approved amount,integer,unit: US dollar
> duration„loan duration,integer,unit: month
> payments,monthly payments,monthly payments,real,unit: month
> status„repayment status,text,"'A' stands for contract finished, no problems; 'B' stands for contract finished, loan not paid; 'C' stands for running contract, OK so far; 'D' stands for running contract, client in debt"
> **Return:** The Loan table manages loan-related data, offering insights into each loan's unique identifier, associated account details, approval dates, amounts, durations, and monthly payments.

### E.3.3 Challenging questions

We discuss three major groups of challenging questions with examples.

The first group contains ambiguous questions. Here are a few examples.

> **Question:** List out the no. of districts that have female average salary is more than 6000 but less than 10000?
> **Reason:** Ambiguous question. 'no. of districts' refers to the district number based on the ground truth. However, Lucy counts the number of districts.

> **Question:** W that the client whose card was opened in 1996/10/21 made?
> **Reason:** Ambiguous question. Lucy filters on 'card issued date', while ground truth filters on 'account opened date'. However, the user is indeed asking about 'card open date' in this question. This issue was also independently observed in [Wretblad et al., 2024a].

The second group contains complex filtering, ordering, and/or formulas to compute. Here are a few examples.

> **Question:** List out the account numbers of clients who are youngest and have highest average salary?
> **Reason:** Phase 3 issue. There are two filtering conditions that have to be applied in order. First, we find the youngest clients, then select the one with the highest average salary among them. Lucy treats these conditions as a conjunction, resulting in an empty output.

> **Question:** List out the account numbers of female clients who are oldest and has lowest average salary, calculate the gap between this lowest average salary with the highest average salary?
> **Reason:** Phase 3 issue. Two filtering conditions are required: first, in descending order, and then in ascending order. However, Lucy fails to perform them in this sequence.

> **Question:** For the client who applied the biggest loan, what was his/her first amount of transaction after opened the account.
> **Reason:** Phase 3 issue. Two filtering conditions are required: first, in ascending order, and then in descending order. However, LUCY fails to perform them in this sequence.

The third group contains questions where the `MatchTables` phase either adds an extra table, or occasionally misses a table or attributes. Here is an example.

> **Question:** How many accounts have an owner disposition and request for a statement to be generated upon a transaction?
> **Reason:** Phase 1 issue. LUCY identifies "Tran" (transaction) as a relevant table, but it is not needed to answer the query.

## E.4  Cloud resources

**Note on the cost of running.** One note here is that GPT and C2Q models are costly to run. For example, in the *Cloud Resources* experiment, the costs are as follows: C2Q costs $15, GPT4 $2, and GPT4EX $5, while LUCY costs $0.5.

