# OpenReview forum: "Lucy: Think and Reason to Solve Text-to-SQL"
_NeurIPS.cc/2024/Conference — Submitted to NeurIPS 2024_

### Official Review · Reviewer_P27x · 2024-06-12

**Soundness:** 2
**Presentation:** 2
**Contribution:** 2
**Rating:** 3
**Confidence:** 4

**Summary:**

This paper addresses the challenge of developing effective LLM-based assistants for querying SQL databases. In this context, users pose questions to a relational database in natural language, and the goal is to generate a SQL query that correctly answers the user's question when executed. The authors focus on overcoming a limitation of current text-to-SQL approaches: the difficulty LLMs face in handling databases with numerous tables and complex relationships, making it hard to determine the necessary table joins for the query.

To tackle this issue, the authors propose a workflow that begins with using the LLM to identify relevant tables and their attributes. In the second step, a constraint satisfaction solver (CSP) is employed to determine the necessary joins while adhering to database constraints. In the third step, a materialized view is created by joining the relevant tables. Finally, this view, combined with the user's question, is used to prompt the LLM to generate the final SQL query.

**Strengths:**

This paper tackles a very relevant practical problem, which is attracting significant attention in both academia and industry. The proposed workflow may add practical value.

**Weaknesses:**

The paper lacks depth, and the writing does not, in my opinion, meet the quality standards required for a venue like NeurIPS. Additionally, several potential limitations of the proposed workflow are not discussed.
	•	Accuracy of the answers is not the only important requirement in generating SQL from text. Database users also expect query generation to be time-efficient. The proposed workflow includes several computationally intensive steps: first, solving an NP-complete problem (CSP), and second, creating a potentially enormous materialized view by joining many tables. I would have expected a discussion on the computational limitations of this approach.
	•	The workflow lacks sufficient precision and clarity. For example, it is unclear whether the final query is expressed with respect to the materialized view as the only table or with respect to the original schema. Additionally, how lookup tables and various schema design patterns are identified in the input database is not well-explained. The authors claim that their approach guarantees the generated query respects database constraints, but this guarantee is not clearly defined. Algorithm 1 is underspecified; at this high level of detail, the algorithm seems redundant and could be subsumed by the text description. The exact SQL fragment covered by this approach is also unclear. While the limitations section mentions that queries requiring the union operator are not supported, it is unclear if other standard SQL constructs are also unsupported.
	•	While relevant related work is cited, the main body of the paper lacks a detailed discussion on the contribution in relation to recent approaches.
	•	The evaluation section is somewhat lacking. The tables are confusing, and it is unclear what each of the rows actually represents.

**Questions:**

Please comment on the limitations concerning unsupported SQL constructs as well as on the computational limitations of the proposed approach.

**Limitations:**

The limitations section mentions some but not all of the relevant limitations of this approach.

---

> ### Author Rebuttal · Authors · 2024-08-06
>
> Thank you for your comments and suggestions!
>
> We will clarify the following points in the paper:
>
> > Efficiency solving an NP-complete problem (CSP)
>
> Solving CSP takes less than a second in all experiments we tried. As the reviewer pointed out, solving CSP is an NP-complete problem. However, modern CSP solvers scale to thousands of variables for industrial benchmarks. In the text2SQL use case, the number of variables is of the order of the number of relevant tables squared. It is a very easy problem for modern solvers and has negligible computational overhead.
>
> Please see more in replies to all reviewers.
>
> > Efficiency of enormous materialized view by joining many tables. It is unclear whether the final query is expressed with respect to the materialized view as the only table or with respect to the original schema.
>
> The view we build is not materialized. Instead, it serves as a sub-query to the final database query. This enables the database query planner to apply standard optimizations such as predicate pushdown to execute the final query efficiently.
>
> > how lookup tables and various schema design patterns are identified in the input database is not well-explained.
>
> Snowflake, many-to-many, star, and other patterns, which we formalize as constraints, are standard database modeling concepts applied by the database designer when creating the database schema. It is therefore straightforward for a domain expert to identify and capture these constraints. In fact, they are often explicitly written down as part of the logical database model (when one exists).
>
> > The authors claim that their approach guarantees the generated query respects database constraints, but this guarantee is not clearly defined.
>
> We guarantee that the generated query satisfies database constraints in the sense that is precisely defined in Section 3.3 C1-C5. As an example, constraint C1 guarantees that when joining a pair of tables with matching primary and foreign keys, the join will be performed using these keys. Constraint C3 guarantees that when joining two tables connected by a many-to-many relation, the join will be implemented as a 3-way join via the auxiliary table.
>
> Our constraint satisfaction problem models all constraints precisely. As the solver is complete and the constraints are hard, we are guaranteed to output a sequence of joins that satisfies these constraints.
>
> > The exact SQL fragment covered by this approach is also unclear. While the limitations section mentions that queries requiring the union operator are not supported, it is unclear if other standard SQL constructs are also unsupported.
>
> The final SQL query consists of the view definition produced with the help of the constraint solver and the query against this view output by the LLM. The former consists of inner and left joins only.  The latter can include arbitrary SQL constructs generated by the LLM.
>
> >  While relevant related work is cited, the main body of the paper lacks a detailed discussion on the contribution in relation to recent approaches.
>
> We will add a comparison with the schema linking method as Reviewer 2 suggested. However, the main contribution is novel (Please see more in the reply to all reviewers.)
>
> > The evaluation section is somewhat lacking. The tables are confusing, and it is unclear what each of the rows actually represents.
>
> We have descriptions of all the metrics that we used for evaluation, including the standard metric from BIRD (ex). We will further clarify row descriptions.

---

> > ### Comment · Reviewer_P27x · 2024-08-12
> > **Author rebuttal**
> >
> > I confirm that I have read the authors' rebuttal and thank the authors for their clarifications. I would like to keep my score.

---

> > > ### Author Response · Authors · 2024-08-13
> > >
> > > Thank you for your comment!
> > >
> > > We would appreciate more feedback from the reviewer on the points not addressed in the rebuttal. We believe we address all technical concerns in the rebuttal.

---

### Official Review · Reviewer_rcsp · 2024-07-12

**Soundness:** 3
**Presentation:** 2
**Contribution:** 2
**Rating:** 6
**Confidence:** 4

**Summary:**

In this paper the authors introduce LUCY, a new LLM based framework for converting text-to-SQL to query databases. Primarily this framework focusses on addressing user queries to databases that contain a large number of tables with complex relations between them.
The core idea of this approach is to decompose the query generation process into distinct stages. LLMs (GPT-4) are utilized for generative tasks such as identifying relevant tables and attributes and generating the SQL query. Meanwhile a deterministic constraint solver (OR-Tools) is employed to map relationships between these elements. In essence LUCY processes a user query through 3 phases namely MatchTables, GenerativeView and QueryView phases. In the MatchTables phase the goal is to identify the relevant tables and attributes. This is accomplished by iteratively prompting a Large Language Model (such as GPT-4) to identify relevant tables and attributes based on the user query and the database model, which includes the schema and an optional list of high-level design patterns. The database model is presented in a hierarchical manner and explored using a breadth-first search approach. Once the relevant relations and attributes are identified a schema graph is constructed and solved using a constrainst solver (i.e to identify the optimal path to join the tables) to build a view in the GenerativeView phase. A LLM is then prompted to generate a SQL query given the summary view and the user query in the QueryView phase. The authors further conduct experiments that demonstrate that the proposed technique achieves a better execution accuracy as compared to the existing state-of-the-art techniques on standard datasets(ACME, BIRD). Furthermore, they also introduce a show large improvements on a new benchmark dataset (Cloud Resources).

**Strengths:**

1. The literature review is comprehensive and the paper does a good job at clearly defining the problem to solve.
2. The novelty of the proposed approach lies in the decomposition of tasks involved in generating SQL queries. By employing LLMs to handle specific subtasks, it effectively circumvents the need for LLMs to perform complex reasoning. A core distinguishing factor from prior research is the use of constraint solvers to identify the relevant paths for joining the identified tables.
3. The authors also demonstrate that the proposed approach achieves a better execution accuracy than the existing SOTA on several benchmarks.

**Weaknesses:**

1. The paper ends abruptly without a clear and comprehensive conclusion. The paper presentation needs improvement in this regard.
2. The authors introduce a new benchmark for evaluation but do not offer sufficient details regarding it. A detailed overview of the queries and an analysis of why the existing SOTA techniques do not perform well on the same could be provided which could greatly inform future work.
3. The practical utility of the proposed technique seems to be limited as each user query requires multiple calls to be made to LLMs thereby entailing both increased latency and cost.
4. The error analysis is not very comprehensive and could be improved. For instance how does this technique fare when the names of entities in database schemas are not semantically meaningful or if there are conflicts in descriptions etc (as is often the case in real-world industrial databases).

**Questions:**

1. For the new benchmark introduced based on the Cloud Resources dataset it is shown that LUCY significantly outperforms the state-of-the-art. What are the queries used in this benchmark? What sets them apart from the standard benchmarks where the execution accuracy of SOTA is comparable/close to LUCY?
2. Has any analysis been done to measure the performance gain vs cost tradeoff when compared to SOTA.
3. What happens when 2 entities in a database have similar descriptions? How sensitive is this technique to having semantically meaningful table names?
4. Is there any mechanism used to handle hallucinations from the MatchTable phase in the GenerativeView Phase?

**Limitations:**

1. As the technique leverages LLMs it seems to be heavily reliant on having semantically meaningful entity names /descriptions
2. The proposed technique seems sensitive to hallucinations as it involves processing a query through multiple LLM phases. The errors in any of the earlier phases would result in it propagating to the next stage. For instance as the authors pointed out if the MatchTables phase produces an extra table this could in turn effect the end output.

---

> ### Author Rebuttal · Authors · 2024-08-06
>
> Thank you for your comments and suggestions!
>
> > For the new benchmark introduced based on the Cloud Resources dataset, it is shown that LUCY significantly outperforms the state-of-the-art. What are the queries used in this benchmark? What sets them apart from the standard benchmarks where the execution accuracy of SOTA is comparable/close to LUCY?
>
> The main distinguishing feature is the complex relationships between tables and queries required to reason about these relationships (more than 6 tables).  Cloud Resources is similar in this respect to ACME Insurance introduced recently by [a]. Lucy works very well on ACME as well (Please see Section 5, ACME insurance). Our results support conclusions from [a]: pure LLM-based approaches do not work on such benchmarks. For example, GPT-4 adds comments like we pointed out in the introduction in most of the outputs:
>
> *'This join may need adjustment based on the actual logic of relating claims to policy coverage details.’*
>
> In summary, these databases are much more complex compared to BIRD databases, including financial and formula1 databases that we tested on, which are the most complex in terms of relationships in BIRD.
>
> #### [a]  Sequeda, D. Allemang, and B. Jacob. A benchmark to understand the role of knowledge graphs on large language model’s accuracy for question answering on enterprise sql databases, 2023
>
> > Has any analysis been done to measure the performance gain vs. cost tradeoff when compared to SOTA?
>
> At the time of submission, Chat2Query is the state-of-the-art method for zero-shot performance. We performed cost analysis for Cloud resources for the benchmarks we reran: Chat2Query costs $15,  while our method costs 50 cents.
>
> A direct performance comparison is hard to perform, as we ran Chat2Query on the cloud service TiDBCloud ( Chat2Query is a closed-source industrial tool) and Lucy runs locally using GPT-4 via API .
>
> Another data point is GPT-4, which costs $2 on the same dataset. The reason that LUCY is cheaper than GPT4 is that we perform iterative traversal on snowflake patterns before feeding them to GPT-4.
> Please see more in the reply to all reviewers.
>
> > What happens when two entities in a database have similar descriptions?
>
> We noticed that Lucy performs well in such cases, given that the descriptions convey meaningful information. Here is an example of three table descriptions from the ACME (insurance) dataset that look similar, at least to non-experts in insurance terminology:
>
> * "Claim Payment":      The amount paid for loss or expense to settle a claim in whole or in part.
> * "Expense Payment": The amount paid for the expenses to settle a claim in whole or in part.
> * "Loss Payment":       The amount paid to claimants to settle a claim.
>
> As Section 5 (ACME insurance results) shows, Lucy produces highly accurate results on this benchmark.
>
> > How sensitive is this technique to having semantically meaningful table names?
>
> We have not experimented with renaming objects. However, we believe that Lucy might be sensitive in the first phase, when detecting relevant objects, if table names are meaningless. Nevertheless, we think all techniques have the same limitation. For example, the idea of evidence in BIRD is to clarify a mismatch between the user's request and how these should be mapped to a SQL query. Consider, for example, a question from the financial dataset.
>
> "query id":  "financial_90":
> "query": "How many accounts that have a region in Prague are eligible for loans?"
> "evidence": "A3 contains the data of region"
>
> The relevant table for this query is DISTRICT that contains meaningless column names like A1, A2, etc., so evidence has to compensate for that.
>
> > Is there any mechanism used to handle hallucinations from the MatchTable phase in the GenerativeView Phase?
>
> MatchTable phase:
>
> There is a chance that MatchTable produces more relevant tables/attributes than necessary. Usually, it is an overapproximation of the true relevant tables/attributes. While additional attributes do not pose an issue, additional tables can indeed be a problem. However, the task of finding relevant tables is much simpler than solving text2SQL as a whole, so experimentally, Lucy performs well.
>
> Also, note that this phase is 100\% hallucination-proof in terms of making up nonexistent tables/attributes (GPT4 is often prone to this type of hallucination).
> * First, we force GPT to output tables/attributes from a predefined set of attributes.
> * Second, we always check that it outputs a subset of predefined tables. Please see D.2.1 promptA and example D2 that show the list of elements GPT has to pick from.
>
> GenerativeView phase:
>
> This phase itself does not introduce additional hallucinations as we do not use LLMs in this phase. Only a constraint solver is used to form a view V.

---

> > ### Comment · Reviewer_rcsp · 2024-08-11
> > **Acknowledgement of rebuttal**
> >
> > I thank the authors for the clarifications. I am keeping my score.

---

### Official Review · Reviewer_ZvFe · 2024-07-12

**Soundness:** 3
**Presentation:** 4
**Contribution:** 3
**Rating:** 4
**Confidence:** 2

**Summary:**

The author proposes a new method, Lucy, designed to handle large databases with complex relationships between objects. Lucy operates through three steps: MatchTables, GenerateView, and QueryView. It first identifies relevant tables and attributes using LLMs, constructs a combined view with an automated reasoner, and generates the final SQL query. Lucy shifts complex reasoning from LLMs to a CSP solver, supporting various database design patterns. Experiments on ACME insurance, Cloud Resources, and the two BIRD databases show that Lucy outperforms other zero-shot text-to-SQL models.

**Strengths:**

- The proposed method offers a fresh perspective on tackling text-to-SQL research with a logical workflow.
- The paper is well-written and easy to follow.

**Weaknesses:**

- I am not convinced by the motivation of zero-shot text-to-SQL with the example of industrial databases having complex relationships. Text-to-SQL systems deployment in real-industry requires high performance. I doubt that people won't be using zero-shot models for real use applications. In KaggleDBQA, it also states "we believe the zero-shot setting is overly-restrictive compared to how text-to-SQL systems are likely to be actually used in practice." I would like to hear the authors' thoughts on this.
- The paper does not appear to be well-grounded in text-to-SQL research. For example, one way to handle complex relationships in text-to-SQL using LLMs is through schema linking. However, the paper does not mention this area of research and instead proposes MatchTables, seemingly ignoring the rich literature of text-to-SQL works. Other approaches include least-to-most prompting attempts in text-to-SQL for task decomposition and Natural SQL for intermediate representation (although it does not handle query nesting). Properly discussing these relevant methods of the proposed method will better situate the work.

**Questions:**

Please address my above concerns. I am willing to raise the score if convinced.

**Limitations:**

The limitations of the work are well-stated.

---

> ### Author Rebuttal · Authors · 2024-08-06
>
> We thank the reviewer for their comments and suggestions!
>
> > Zero-shot vs multi-shot
>
> In large industrial databases with complex relationships, using multi-shot will not improve accuracy with respect to database constraints. There are two reasons for that:
>
> * First, the structure of the database is very complex, with easily hundreds of tables.
> * Second, user questions are very diverse, as the user can query about any aspect of their database.
>
> Hence, getting predefined or automatically generated examples that allow LLMs to capture dependencies between 6 or more tables to answer a user query is as hard as answering the original user request.
>
> > Schema linking
>
>
> We appreciate the reviewer pointing us to the schema linking work. This is indeed a similar approach with a similar goal. We will cite the literature on this topic. The main difference in our approach is that we can handle database structures, like snowflake efficiently. However, we can definitely borrow some ideas from the literature to improve this phase, e.g., ideas from latest work that uses linking schema [a]
>
> However, we would like to highlight that our main contribution -- *the separation of responsibilities between LLMs and automated reasoners* -- is new to the best of our knowledge.
>
> #### [a] Tapilot-Crossing: Benchmarking and Evolving LLMs Towards Interactive Data Analysis Agents Jinyang Li, Nan Huo, Yan Gao, Jiayi Shi, Yingxiu Zhao, Ge Qu, Yurong Wu, Chenhao Ma, Jian-Guang Lou, Reynold Cheng https://arxiv.org/abs/2403.05307

---

> > ### Author Response · Authors · 2024-08-11
> >
> > Dear Reviewer, we hope our rebuttal addressed addressed your concerns.  We are happy to answer any additional questions.

---

> > > ### Comment · Reviewer_ZvFe · 2024-08-14
> > >
> > > Thank you for your response. I disagree with the statement that using multi-shot examples will not improve accuracy with respect to database constraints. I argue that multi-shot demonstrations can enhance this aspect, especially if the exemplars are from the same database as the input query being processed. While the motivation is good, the claims in this work do not seem to be well-grounded in the existing literature on text-to-SQL, so I am inclined to maintain my score.

---

> > > > ### Author Response · Authors · 2024-08-14
> > > >
> > > > Thank you for your comments!
> > > >
> > > > As we mentioned in the rebuttal, implementing multi-shot is highly non-trivial for complex databases and diverse queries that involve 6 or more tables. In the paper, we demonstrated a significant improvement compared to the current best methods using zero-shot.
> > > >
> > > > It might be possible to use multi-shot to get even better results, but it is a subject of future research.

---

### Official Review · Reviewer_YkDn · 2024-07-15

**Soundness:** 1
**Presentation:** 1
**Contribution:** 2
**Rating:** 3
**Confidence:** 4

**Summary:**

The paper introduces Lucy, a framework for solving Text2SQL by LLMs, particularly for complex enterprise databases. Lucy leverages LLMs' understanding and reasoning capabilities to handle intricate database relationships and constraints. The framework operates in three phases: identifying relevant tables and attributes (MatchTables), constructing a view through constraint reasoning (GenerateView), and generating the final SQL query (QueryView). The empirical studies show Lucy achieves performance improvements on several zero-shot Text2SQL tasks.

**Strengths:**

Text2SQL is an essential problem in commercial scenarios.

**Weaknesses:**

The draft seems far from complete, so leave some high-level suggestions.
1. Make the title, abstract, and introduction more concrete. It is hard to tell the contribution or uniqueness of this work among other papers about Text2SQL by LLMs.
2. Survey related works and clearly state the contribution/novelty of the proposed method against others.
3. Define the terminologies or abbreviations before their first appearance.
4. Make the draft concise by removing unnecessary content. For example, the first challenge introduced in Motivation section is not relevant to this work.
5. The empirical studies could be more convincing by following others' evaluation protocols, such as BIRD.
6. Lack of comparison to other competitors.
7. The figures, tables, and their captions should be self-explanatory.

**Questions:**

1. What is the key difference between the proposed framework and competitors (e.g., MCS-SQL, MAC-SQL, Chat2Query)?
2. How does the proposed method perform if evaluated by the same protocol as the BIRD-SQL leaderboard?

**Limitations:**

There is a discussion about the limitation, though the first limitation seems too broad and unnecessary.

---

> ### Author Rebuttal · Authors · 2024-08-06
>
> Thank you for your comments and suggestions!
>
>
> > What is the key difference between the proposed framework and competitors (e.g., MCS-SQL, MAC-SQL, Chat2Query)?
>
> The key difference is that all text-to-SQL methods rely on LLMs to reason about database constraints. These constraints are hard constraints that must be enforced in any valid query. LLMs have been shown to be weak in reasoning about hard constraints. In our work, we use automated reasoning tools to enforce database constraints. *We emphasize that none of the existing techniques have such capabilities.*
>
> > How does the proposed method perform if evaluated by the same protocol as the BIRD-SQL leaderboard?
>
> We use the exact same protocol as the BIRD leaderboard (e.g. Leaderboard - Execution Accuracy (EX)). Please row 'ex' in Tables 2-5. Other metrics are used in addition to 'ex'.
>
> > There is a discussion about the limitation, though the first limitation seems too broad and unnecessary.
>
> We respectfully disagree with the reviewer regarding the first limitation. It is a critical capability of our method that we generate queries that satisfy database pattern constraints. While it is not possible for any method to provide full guarantees that a SQL query answers the user’s question at the moment, our approach is a step forward in the direction to produce correct queries.
>
> > Presentation comments
>
> We will improve presentation taking into accounts reviewer's comments.

---

> > ### Comment · Reviewer_YkDn · 2024-08-09
> >
> > Thank the authors for the response. I've recognized the contribution, so I adjust my score from 2 to 3 accordingly. However, as others and my comments mentioned, the presentation needs improvement to match the quality of the conference. The related work of Text2SQL and automated reasoning should also be discussed more comprehensively.

---

> > > ### Author Response · Authors · 2024-08-11
> > >
> > > Thank you for adjusting the score. We are puzzled by the final reject, given that
> > >
> > > - Our detailed evaluation demonstrates that this work significantly improves over the current state of the art in zero shot text2sql
> > > - We have satisfactorily addressed reviewer's concerns (as indicated in the reviewer's last comment)
> > > - Presentation-related issues pointed out by reviewers can be easily addressed in the final version of the paper, and we outlined our suggested improvements in the rebuttal
> > > - While we missed related work on schema linking, it does not affect the novelty of this work and can easily be incorporated in the related work section in the final version of the paper.
> > >
> > > We therefore believe that the reject score is not justified given the novelty of the proposed method and the significance of the results

---

### Author Rebuttal · Authors · 2024-08-06

We thank you for your comments for their comments! We would like to clarify the following important points:

[Main contribution] Our primary contribution is the **elimination of the weakest point in LLM-based text-to-SQL methods: reasoning about database constraints**. We propose utilizing powerful automated reasoner tools to perform such reasoning. To the best of our knowledge, none of the existing methods provide this capability, which is crucial for large databases with complex relationships.


[Database patterns] In this work, we assume that the text-to-SQL developer has to specify database patterns, like many-to-many, snowflake, lookup, etc. It is a reasonable assumption for many SaaS services where the service provider runs data analytics on behalf of customers, for example.

[Performance overhead] Our method does require solving CSP and building an intermediate view table. However, the overhead of both these steps is negligible in practice.

* Solving CSP is very fast because it is a small and underconstrained model, which can be easily handled by any modern solver (in less than a second).
* The view we build is not materialized. Instead, it serves as a sub-query to the final database query. This enables the database query planner to apply standard optimizations such as predicate pushdown to execute the final query efficiently.

*We would like to emphasize that **formal reasoning about a set of logical constraints will remain beyond reach of LLMs in the near future**. For example, there is no evidence to suggest that LLMs can replace automated reasoners, such as SAT, SMT, and CSP solvers. Therefore, logical reasoning must be performed by specialized solvers for text2sql tools to be useful in practice.*

---

### Decision · Program_Chairs · 2024-09-25

**Decision:**

Reject

**Comment:**

While the paper presents a novel approach to an important problem, the significant weaknesses in presentation, clarity, and some concerns about the technical depth and practical applicability of the method outweigh the strengths in the area of database constraint reasoning. Even though the authors argue that those issues can mostly be fixed with moderate effort, another round of reviews would be needed to assess whether such modifications meet the very high quality bar of NeurIPS. Given these issues, I recommend rejecting the paper.